# IKEA-Manual: Seeing Shape Assembly Step by Step

**Ruocheng Wang**
Stanford University

**Yunzhi Zhang**
Stanford University

**Jiayuan Mao**
MIT

**Ran Zhang**
Autodesk

**Chin-Yi Cheng**[*]
Google Research

**Jiajun Wu**
Stanford University

## Abstract

Human-designed visual manuals are crucial components in shape assembly activities. They provide step-by-step guidance on how we should move and connect different parts in a convenient and physically-realizable way. While there has been an ongoing effort in building agents that perform assembly tasks, the information in human-design manuals has been largely overlooked. We identify that this is due to 1) a lack of realistic 3D assembly objects that have paired manuals and 2) the difficulty of extracting structured information from purely image-based manuals. Motivated by this observation, we present IKEA-Manual, a dataset consisting of 102 IKEA objects paired with assembly manuals. We provide fine-grained annotations on the IKEA objects and assembly manuals, including decomposed assembly parts, assembly plans, manual segmentation, and 2D-3D correspondence between 3D parts and visual manuals. We illustrate the broad application of our dataset on four tasks related to shape assembly: assembly plan generation, part segmentation, pose estimation and 3D part assembly.

## 1 Introduction

Shape assembly is a common task in our daily lives. Given a list of parts, we sequentially move and connect them to obtain the target shape—a LEGO toy, an appliance, or an IKEA chair. Due to the long-horizon nature of the task, we often heavily rely on visual manuals that provide step-by-step guidance during the assembly process. Previous studies [1, 2] have shown that these carefully designed manuals decompose the assembly task into a series of simple actions while taking multiple factors into account, including readability, assembly difficulty, and physical feasibility.

While there is a growing interest in building autonomous models that tackle the shape assembly task in the vision [3, 4, 5] and robotics communities [6, 7, 8], the information in human-designed manuals is largely overlooked. We identify two reasons: first, most works leverage shapes from synthetic datasets such as PartNet [9] as the assembly targets, which do not have corresponding human-designed manuals; second, directly leveraging the information from visual manuals is challenging: these manuals are delivered in plain unstructured images, guiding humans via visual elements including arrows and the 2D projections of 3D shapes. While humans can effortlessly understand these visual elements, this is not the case for machines.

Motivated by these observations, our goal is to create a dataset for studying shape assembly that contains realistic 3D assembly objects paired with human-designed manuals, where the manuals are parsed in a structured format. Towards this end, we introduce IKEA-Manual, a dataset of 3D IKEA models and human-designed visual manuals with diverse types of annotations, as shown in Fig. 1. IKEA furniture is well known for being manufactured with assembly parts and visual manuals

---

[*]Work done when working at Autodesk AI Lab.

36th Conference on Neural Information Processing Systems (NeurIPS 2022) Track on Datasets and Benchmarks.

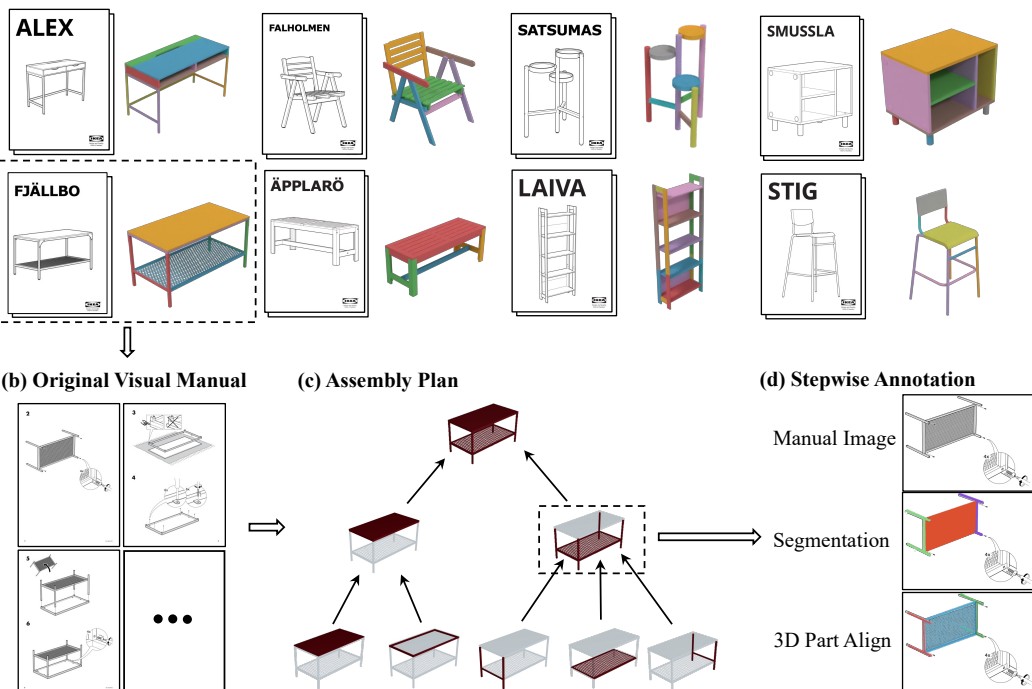

Figure 1: We present IKEA-Manual, a dataset for step-by-step understanding of shape assembly from 3D models and human-designed visual manuals. (a) IKEA-Manual contains 102 3D IKEA objects paired with human-designed visual manuals, where each object is decomposed into primitive assembly parts that match manuals shown in different colors. (b) The original IKEA manuals provide step-by-step guidance on the assembly process by showing images of how parts are connected. (c) We extract a high-level, tree-structured assembly plan from the visual manual, specifying how parts are connected during the assembly process. (d) For each step, we provide dense visual annotation such as 2D part segmentation and 2D-3D correspondence between 2D manual images and 3D parts.

for customers to assemble by themselves. Each IKEA manual contains information about how to move 3D parts to desired poses by projecting them onto the manual image step by step. A number of datasets [10, 11, 6] include 3D shapes of IKEA furniture, but the manual information is not included.

Specifically, we collect 102 3D objects of IKEA furniture from existing datasets and online repositories, paired with corresponding manuals from IKEA's official website. For each object, we decompose its 3D model into assembly parts that match the assembly manual and annotate their connection relationships. We annotate assembly manuals by first recording which parts are connected in each step, and then manually segmenting out the projection of parts on the manual images. Finally, we annotate keypoints on the 3D assembly parts and their 2D projections on the manual image, and use them to compute the 3D poses of parts and build 2D-3D correspondences.

We demonstrate the usefulness of IKEA-Manual on four applications. First, we propose a novel assembly plan generation task, where models are asked to generate an assembly plan tree given a shape and its part decomposition. The task not only helps robots to plan for the assembly task in a high level, but also facilitates the laborious process of designing assembly manuals. Second, we use our dataset for part segmentation, where the goal is to identify image segments that correspond to input 3D parts. This is a prerequisite for building agents that understand and translate visual manuals into machine-interpretable instructions. Then we demonstrate IKEA-Manual can also be used to benchmark the task of part-conditioned pose estimation, where we want to predict the poses for 3D shapes in manual images. Finally, we use shapes from our dataset to evaluate the part assembly task [4]: given a list of parts, models need to predict their final poses such that all parts are assembled into a single 3D shape.

In summary, we introduce IKEA-Manual, a dataset for understanding shape assembly from 3D shapes and human-designed visual manuals. It contains 102 3D objects of IKEA furniture paired with visual

| Annotation Type | | PartNet [9] | Pix3D [11] | Fusion 360 Gallery [5] | AutoMate [12] | IKEA Furniture [6] | IKEA OS [13] | IKEA-Manual |
|---|---|---|---|---|---|---|---|---|
| Shape | Part Decomposition | ✓ | ✗ | ✓ | ✓ | ✓ | ✓ | ✓ |
| | Real-World-Aligned Parts | ✗ | ✗ | ✓ | ✓ | ✗ | ✓ | ✓ |
| Manual | Paired Assembly Manual | ✗ | ✗ | ✗ | ✗ | ✗ | ✓ | ✓ |
| | Assembly Plan | ✗ | ✗ | ✗ | ✗ | ✗ | ✗ | ✓ |
| | Manual Segmentation | ✗ | ✗ | ✗ | ✗ | ✗ | ✗ | ✓ |
| | 2D-3D Correspondence | ✗ | ✓ | ✗ | ✗ | ✗ | ✗ | ✓ |

Table 1: A comparison of IKEA-Manual and other representative IKEA or part-level shape datasets. IKEA-Manual consists of real-world IKEA objects with both 3D part decomposition and human-design manuals with dense annotations. In comparison, previous datasets lack either part decomposition that aligns with real-world IKEA objects, or structured information of assembly manuals.

assembly manuals. We provide dense annotations on the 3D models and assembly manuals, as well as correspondences between 3D assembly parts and their projections in the visual manuals. We conduct four experiments in manual plan generation, part segmentation, pose estimation and 3D part assembly to illustrate the usefulness of our dataset.

## 2 Related Work

**3D part datasets and IKEA datasets.** We summarized the comparison of IKEA-Manual and other representative 3D part datasets and IKEA datasets in Table 1. A number of 3D part datasets have been proposed in the last decade. Previous works mainly focus on 3D shape understanding [14, 15, 16, 9], where 3D models of daily objects are segmented into semantic classes. Recently, researchers have proposed a number of works that contain parametric 3D models of more diverse objects [17, 18], where they are decomposed into geometric primitives that reflect the modeling processes of creators. These datasets are mainly used for low-level geometric shape analysis. AutoMate [12] was created for analyzing CAD assemblies, but focuses on modeling the individual mating relationship between mechanical parts without considering target shapes. Fusion 360 Gallery [19, 5] is similar to AutoMate, but not constrained to pairwise assembly. Objects are decomposed to assembly parts that match the manufacturing scenario. It also presents a tree-structured annotation, but represents the semantic hierarchy of shapes rather than guiding the assembly process. Furthermore, most shapes in Fusion 360 are not designed for humans to assemble and thus do not have corresponding manuals. IKEA-Manual, in contrast, focuses on IKEA assembly objects paired with assembly manuals.

On the other hand, IKEA objects have also been leveraged in other domains. IKEA [10] and Pix3D [11] pair 3D IKEA objects with 2D real photos to study single-view pose estimation and 3D reconstruction tasks on the object level. They do not provide any part-level annotation. IKEA Furniture [6] leverages IKEA objects to study shape assembly problems in a simulated environment. It also provides assembly parts for the 3D shapes, though the parts are often not aligned with the parts in manuals. The IKEA Object State Dataset is the most similar to our dataset, as it contains decomposition of IKEA objects paired with manuals. But there are only 5 objects in the dataset and the manuals do not have any form of annotations, which prevents machines from leveraging their assembly information. IKEA-Manual contains 102 objects of greater diversity paired with manuals parsed in machine-interpretable form.

**Shape assembly.** Shape assembly has been studied from various perspectives. A line of research focuses on learning to predict the poses of parts that assemble the target shape [3, 4, 5, 20]. These works use shapes from synthetic datasets such as PartNet [9] and do not consider the motion planning of moving and mating the part. On the other hand, IKEA-Manual provides 3D objects with part decomposition that align with real-world IKEA, which helps evaluate the performance of such models in practical scenarios. Several simulation environments have been proposed to train agents for assembly tasks by reinforcement learning algorithms [21, 6, 22, 23, 24, 8], where either assembly parts and target shapes are greatly simplified, or a hand-engineered dense reward is used to solve the target task. IKEA-Manual may facilitate these agents by providing guidance via the assembly plan.

**Parsing human-designed diagrams.** Humans commonly use diagrams to communicate various types of concepts and information [1]. Building agents that automatically extract information from these diagrams has been studied in the domain of engineering drawings [25], cartographic road maps [26] and sewing patterns [27]. Shao et al. [28] proposed a method to parse assembly IKEA manuals automatically into structured assembly instructions, but it requires specialized data labeling

and has strict constraints on the geometry of 3D parts. In our work, we create tools to manually label manual information to cover more diverse IKEA objects. IKEA-Manual can be used for evaluating models that parse assembly manuals.

# 3 IKEA-Manual

In this section, we provide an overview of different types of annotations in IKEA-Manual, followed by an analysis of our datasets.

## 3.1 IKEA Shape Assembly Guided by Manuals

We begin our section by first providing some background of IKEA shape assembly guided by visual manuals. An IKEA manual decomposes the assembly task into multiple steps. Initially, we are given a list of primitive assembly parts. In each step, we move and connect some parts moved to form new compositional parts, and we complete every step sequentially until the final target shape is assembled. We will use *assembly parts* to refer to both the primitive assembly parts before we begin the assembly and the compositional parts during the assembly.

In each page of an IKEA manual, there are multiple types of visual elements that guide the assembly process, presented as vector graphics, as shown in Fig. 2. The most important information is the projection of the parts involved, which describes the pair of parts that are connected and their desired relative poses. Connectors like screws and hinges, which are used to fasten the parts, are also presented in manuals. Similar to previous work [6], we will ignore them in our 3D shapes and visual manuals, focusing on the connection relationship between parts, as connectors are challenging to model and typically absent in the collected 3D models. We also treat other elements like detailed instructions as part of the background and focus on the assembly parts information.

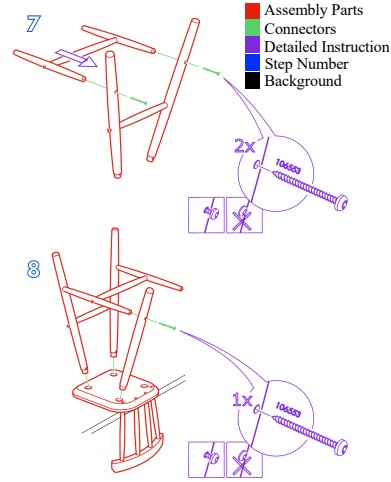

Figure 2: Sample (colored) page of an IKEA manual, which has assembly information of two consecutive steps. Involved parts and their relative poses are presented by their projection on the manual images, which are the main focus of our annotation. Other visual elements including connectors are treated as background in IKEA-Manual.

## 3.2 Data Collection

We first crawl a list of IKEA manuals from the IKEA official website [29]. For each manual, we search for its corresponding 3D model from multiple sources: 1) previous 3D shape datasets that contain IKEA objects like IKEA [10], Pix3D [11], and the IKEA Object State Dataset [13]; 2) online repositories such as 3D Warehouse [30] and Polantis [31]. Except for the IKEA Object State Dataset, shapes from other sources do not come with assembly parts. We do not use shapes from the IKEA furniture assembly environment [6], because many of its shapes do not have corresponding manuals. Finally, we exclude manuals for which we cannot find accurate corresponding shapes, resulting in 102 pairs of shapes and manuals.

## 3.3 Annotation Types

Given the collected IKEA objects with 3D models and corresponding manuals, we include a wide range of annotations on the 3D models, 2D manuals, and their 2D-3D correspondence. First, we provide annotations on the 3D models, as illustrated in Fig. 4:

**Assembly part decomposition.** Realistic decomposition of IKEA shapes into assembly parts is crucial for aligning the parts with guidance from visual manuals. It enables a number of tasks studying the relationship between assembled shapes and visual manuals, such as part segmentation and pose estimation. At the same time, assembly parts can be used to evaluate performance on assembly tasks [3, 4] that receive such decomposition as the input. Therefore, we manually separate the 3D model into primitive assembly parts according to the manuals.

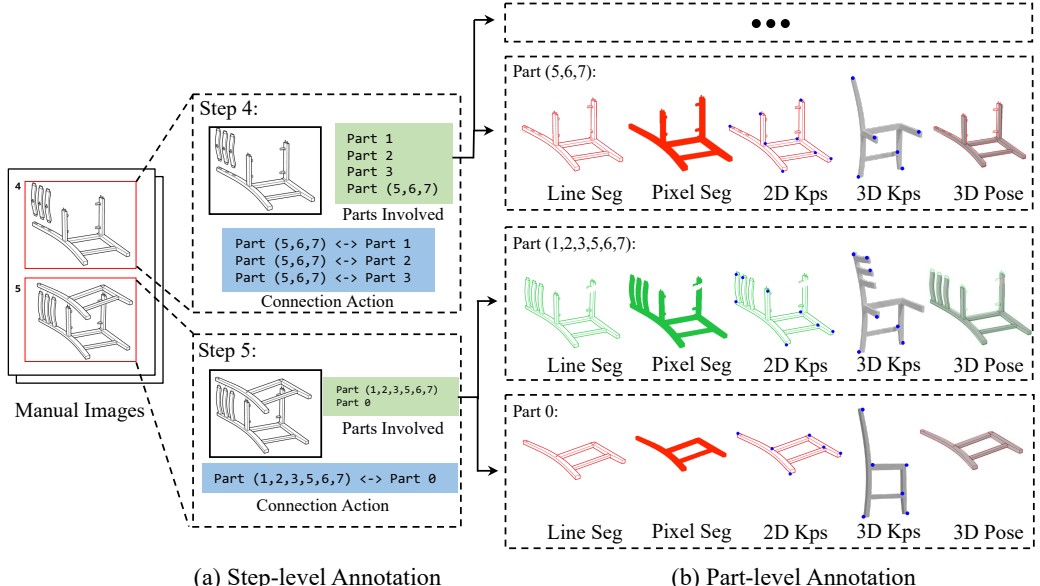

| (a) Step-level Annotation | (b) Part-level Annotation |

Figure 3: Workflow for annotating manual-related information. We first identify assembly steps in the manual. Then we record parts involved in each step and their connection relationships. Finally, we annotate segmentation and keypoints information for each part. Kps is abbrieviated for keypoints.

**Part connection relation.** Knowledge of which primitive assembly parts are finally connected is useful for studying the shape assembly process. We record which parts are directly connected in the final assembled object.

**Part geometrical equivalence relation.** Due to the prevalent symmetry of furniture objects, different primitive assembly parts in IKEA-Manual can share the same canonical geometric shape, which leads to ambiguous input/outputs that hamper training and evaluation of tasks involving 3D parts. Therefore we also include a relation annotation which indicates whether any two parts are geometrically equivalent regardless of their poses.

In addition to 3D models, we also provide rich annotations on visual manuals. The workflow for annotating manual-related information is shown in Fig. 3. We detect areas in the manual image that correspond to each step in the assembly process, excluding steps that only involve connectors, and provide the following fined-grained annotations:

**High-level assembly plan.** We first record the pair of parts that are connected in each step. Combining this information across steps can yield a tree structure of the assembly plan as shown in Fig. 1c . Each non-leaf node represents the target part in a step, and its children define the assembly parts needed to assemble the target part from previous steps. This can be treated as high-level planning information specified by human designers to describe the decomposed assembly process. The representation is structured and machine-friendly to be leveraged in robot assembly [7]. On the other hand, it can also serve as a benchmark for evaluating the assembly plan generation task, which is valuable in computer-aided-design [32, 33, 2] and robot planning [23, 24, 23].

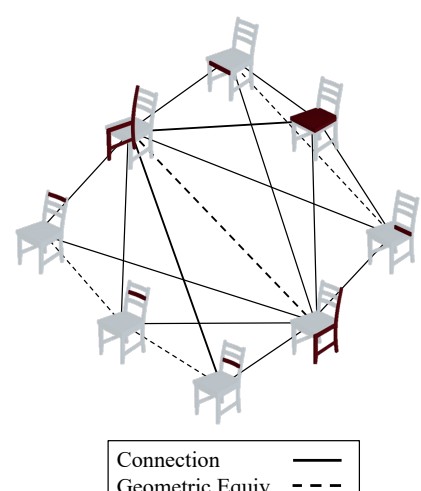

Figure 4: Annotated connection and geometric equivalence relationships between parts.

**Visual manual segmentation.** Identifying the location of assembly parts on the manual image is an important first step toward manual understanding. Therefore, we include two types of instance segmentation of parts involved in each step. As original

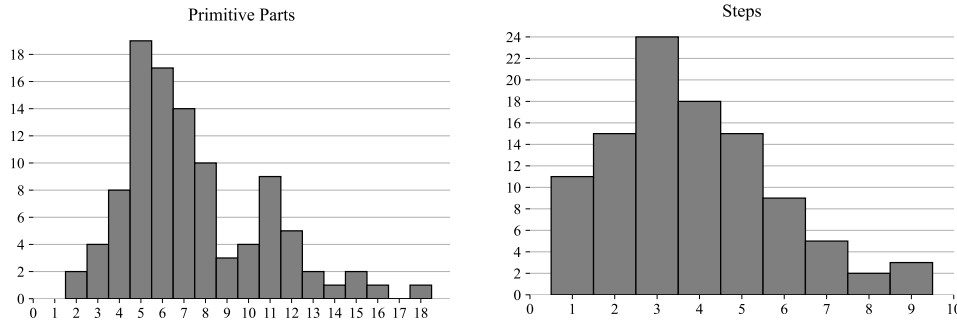

(a) Distribution of # of primitive parts across different objects.  (b) Distribution of # of steps across different objects.

Figure 5: Basic statistics of IKEA-Manual.

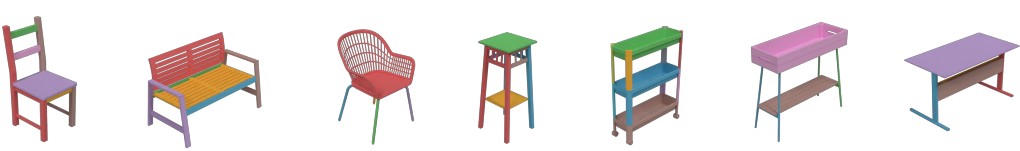

Figure 6: Examples of 3D IKEA objects with assembly parts in different colors. IKEA-Manual have a divers set of object shapes as well as primitive assembly parts.

IKEA manuals are in vector graphics format, which consists of vector graphic primitives such as lines and polygons, we directly assign each primitive to either the corresponding 3D part or the background. This type of annotation on line segments makes IKEA-Manual suitable for evaluation on tasks related to sketch segmentation [34]. To create a better benchmark for pixel-based segmentation, we also annotated a pixel-based segmentation where the inner area of each part is also considered in the mask.

**2D-3D alignment between parts and manuals.** Building alignment between 3D parts and their 2D projections on the manual image enables the reader to infer the relative pose between connected parts. We achieve such alignment by annotating corresponding keypoints on 3D parts and 2D manuals for every part in every step. Once we have these keypoints, we use the Efficient Perspective-n-Point algorithm [35] to compute their poses on the manual image. Combined with the annotation of assembly parts, this annotation can be leveraged in pose estimation [36, 37] and single-view 3D reconstruction [38, 39] tasks.

### 3.4 Dataset Analysis

In total, IKEA-Manual includes 393 steps from 102 IKEA objects, along with 1056 part masks and 3D poses for primitive and compositional assembly parts in every step. It includes 57 chairs, 19 tables, 8 benches, 4 desks, 3 shelves, and 8 objects from other categories like plant stands and carts. Chairs comprise a significant percentage of the dataset, as their 3D models are most abundant in online repositories [16, 11]. Fig. 5 shows the distribution of primitive parts and steps in IKEA-Manual. Our dataset covers a diverse type of assembly part decomposition and assembly plans.

We also present examples of shapes in Fig. 6, with primitive assembly parts painted in different colors. The parts cover a diverse range of geometries. Most notably, this part decomposition is different from previous datasets as an assembly part may not have a single semantic class. For example, in the leftmost shape in Fig. 6, the red assembly part belongs to both the chair back and legs. The purple part in the second shape belongs to both the bench arm and bench leg class. This makes IKEA-Manual a unique resource for shape assembly tasks that aligns better with practical scenarios.

## 4 Application

Our diverse set of annotations in IKEA-Manual enables several downstream applications related to shape assembly. In this section, we demonstrate this through three applications: manual generation, manual interpretation and assembly part analysis. First, we formulate the task of assembly plan

generation, and present an evaluation protocol based on our dataset. Second, we consider a part-conditional segmentation task on manual images, which can be treated as a prerequisite for artificial agents to interpret visual assembly manuals. Finally, we evaluate pretrained models on the task of 3D part assembly to demonstrate the geometric diversity of assembly parts in our dataset.

## 4.1 Assembly Plan Generation

When assembling a target shape, we need to plan for the order of assembly actions. This has been a long-standing problem in computer-aided design [32, 33, 2] and task planning [23, 24, 23]. Here, we are interested in whether we can build algorithms that automatically generate assembly plans. There are multiple factors to evaluate an assembly plan, such as clarity and effectiveness [33], which are hard to define by objective metrics. Previous works often evaluate the quality of generated plans by case studies or human subject experiments [33], IKEA-Manual suggests an automatic, quantitative evaluation protocol based on the assembly plans from IKEA manuals that are professionally designed.

**Assembly tree representation.** We show the idea of representing an assembly plan as a tree in Fig. 1a. Formally, given a target assembly shape with parts $\{P_1, P_2, \cdots, P_n\}$, we say a directed acyclic graph $T$ is an **assembly tree** of $S$ when the following properties are satisfied:

- Each node $N$ corresponds to a non-empty subset of the parts: $\varnothing \subsetneq S(N) \subseteq \{P_1, P_2, \cdots, P_n\}$.
- The root node corresponds to the final assembled shape: $S(N_{root}) = \{P_1, P_2, \cdots, P_n\}$.
- The child nodes for any node are a partition of it, namely $S(N) = \bigsqcup_{N' \in \text{Children}(N)} S(N')$.

Semantically, each leaf node corresponds to a single primitive assembly part and each non-leaf node corresponds to the assembled result from connecting multiple parts (that may either be primitive or assembled) in a step. The whole tree represents an assembly plan. Note that such a tree can also encode the order of the assembly steps: we visit every non-leaf node by post-order traversal and treat the visit order as the assembly order.

We note that the tree representation is not equivalent to the assembly information conveyed in a visual manual – the way parts are moved to be connected is ignored and we do not record how the connection actions are projected to the manual image. However, based on our experiments, predicting such simplified information is already challenging. Therefore, we believe the assembly plan generation task can be treated as a prerequisite for the assembly manual generation task.

**Problem formulation.** Given a target assembly shape represented by a list of primitive assembly parts $\{P_1, P_2, \cdots, P_n\}$, our goal is to predict a valid assembly tree $T$, which satisfies the properties mentioned in the previous paragraph. In our experiment, each assembly part is represented by a point cloud $P_i \in \mathbb{R}^{d_{pc} \times 3}, \forall 1 \le i \le n$.

**Metrics.** Motivated by the dependency tree evaluation in natural language processing [40], we propose to use precision/recall-based metrics that compare the nodes between predicted and ground-truth assembly trees. Specifically, we first use two different criteria to find matching pairs of nodes:

- Simple matching: We match two non-leaf nodes as long as they correspond to the same set of primitive parts. This metric measures the number of correct compositional assembly parts with respect to the ground-truth assembly tree, without considering how they are assembled.
- Hard matching: We match two non-leaf nodes if they correspond to the same set of primitive parts, and moreover, their children nodes are matched by the simple criterion. This criterion can measure the number of correct intermediate assembly actions with repsect to the ground-truth assembly tree, without considering their orders.

We then compute the following metrics for each criterion:

$$\text{Precision} = \frac{\text{\# of matched non-leaf nodes}}{\text{\# of predicted non-leaf nodes}}, \quad \text{Recall} = \frac{\text{\# of matched non-leaf nodes}}{\text{\# of ground-truth non-leaf nodes}}.$$

**Baselines.** We consider two heuristic baselines for the manual plan generation task: The first baseline, *SingleStep*, simply connects all primitive parts in a single step, which corresponds to a single parent node with $n$ leaf nodes for each primitive part. Motivated by the observation that geometrically similar parts are prone to be assembled in the same steps (e.g., for the legs of a chair), we implement a clustering-based baseline *GeoCluster*. It first uses a pretrained DGCNN [41] to extract geometric features for each primitive part, and then recursively groups geometrically similar parts into one step. More details are in the supplementary materials.

|            | Simple Matching | | | Hard Matching | | |
|------------|------|------|------|------|------|------|
|            | Prec | Rec | F1 | Prec | Rec | F1 |
| SingleStep | **100** | 35.77 | **48.64** | 10.78 | 10.78 | 10.78 |
| GeoCluster | 44.90 | **48.46** | 43.53 | **16.54** | **16.50** | **16.30** |

Table 2: Quantitative results of the assembly plan generation. GeoCluster is better on most metrics.

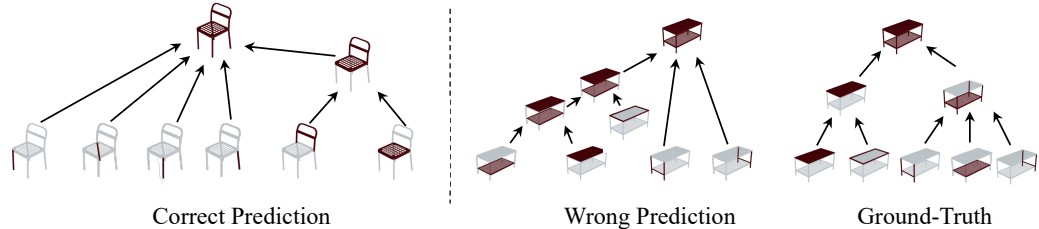

Correct Prediction          Wrong Prediction          Ground-Truth

Figure 7: Visualization of assembly trees generated by GeoCluster. As GeoCluster only considers the geometric similarity between parts, it fails to take connection account into consideration.

**Results.** Quantitative results are shown in Table 2. Neither of the two baselines perform well on the task, which demonstrates its difficulty. SingleStep achieves perfect precision because it will not output any nodes apart from the root node, which is always present. We provide success and failure cases in Fig. 7. We can see that GeoCluster can handle simple cases where geometric similarity is the main factor of assembly planning, but fails to consider the connection constraints between parts.

## 4.2   Part-Conditioned Manual Segmentation

To understand the assembly information of a visual manual, an agent must first locate the assembly parts illustrated in the manual and then infer their 3D positions by finding 2D-3D correspondences. Here we focus on the part-conditioned manual segmentation task, a prerequisite for understanding visual manuals: given a list of parts in their canonical space, and a manual image that contains their 2D projections, we want to predict the segmentation mask for each part, as shown in Fig. 8a.

Specifically, we crop out the parts' projections on manual images individually for each step. Then we treat the resulting image along with the point clouds of involved parts as the input. Because no previous segmentation models have been trained on real-world manual images, some fine-tuning is required. Thus, we split IKEA-Manual into 353 train examples and 40 test examples so that we can finetune the pretrained segmentation model. We leverage the segmentation model in [3], which is a U-Net [42] architecture conditioned on geometric features from PointNet [43]. We finetune all layers of its checkpoint pretrained on a chair dataset from PartNet [9] with learning rate $10^{-4}$ and obtained an IoU of 25.31, which shows that Li et al. [3] struggles on the task. We provide visualizations in Fig. 8b. Li et al. [3] has difficulty predicting masks of more complex parts.

## 4.3   Part-Conditioned Pose Estimation

In this section, we demonstrate that our dataset can also be leveraged to benchmark the part-conditioned pose estimation task: given a 3D part and an manual image that contains its 2D projection, the model needs to predict the rotation of the 3D part.

**Setup.** Similar to the task of part-conditioned manual segmentation, we also crop out the projections of parts on the manual image. We follow the setting of Xiao et al. [44] where only one part is involved for a single example. There are 1056 examples in total, which is randomly split into 844 for training, 105 for validation and 107 for testing. We evaluate three baselines for this task. The first baseline simply outputs a random rotation regardless of the input. This can be treated as the lower-bound of all the metrics. The second baseline is Xiao et al. [44], a learning model that leverages ResNet [45] and PointNet [46]. For this baseline, to prevent distractions of other parts in the same manual image, we multiply the image with the mask of the input part for each example. The third baseline is SoftRas [47], which is a differentiable render that can be leveraged to optimize for the pose from

| | Rotation Error ↓ | Rotation Accuracy ↑ | CD ↓ | CA ↑ | Inference Time (s) ↓ |
|---|---|---|---|---|---|
| Random | 103.49 | 3.05 | 0.1344 | 0.28 | N/A |
| Xiao et al. [44] | 82.79 | 20.56 | 0.0984 | 49.53 | $< 10^{-5}$ |
| SoftRas [47] | **64.81** | **24.30** | **0.0976** | **51.4** | 243 |

Table 3: Quantitative results for the part-conditioned pose estimation task. Xiao et al. [44] and SoftRas both achieve performance higher than chance. SoftRas is slightly better but also significantly slower due to the process of optimization during inference.

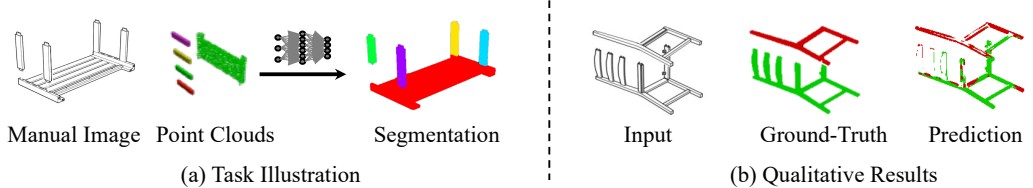

(a) Task Illustration         (b) Qualitative Results

Figure 8: Experiment on the part-conditioned manual segmentation task. The baseline model struggles to predict the segmentation masks of parts assembled from multiple primitive parts.

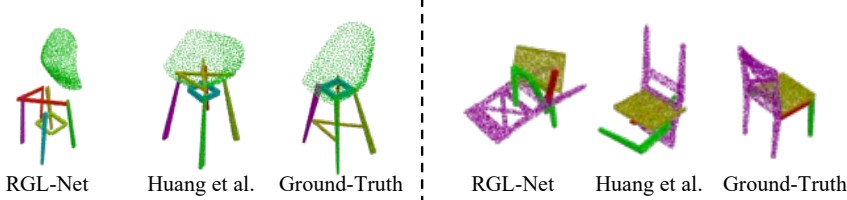

RGL-Net    Huang et al.    Ground-Truth      RGL-Net    Huang et al.    Ground-Truth

Figure 9: Qualitative results for the part assembly task. The model assembles chair-like shapes when parts are simple and semantically meaningful, but fails with more complex parts.

masks in an analysis-by-synthesis fashion. To avoid local minima, we perform optimization from multiple initializations and select the solution with minimum mean squared loss.

**Metrics.** We first adopt the rotation error and rotation accuracy metrics from Xiao et al. [44]. The rotation error directly computes the relative rotation between the ground-truth rotation and predicted rotation for each example in degree unit and take the median over the test set. The rotation accuracy metric computes the percentage of examples whose rotation error is less than $30°$. Because there are a lot of symmetric parts in IKEA-Manual, we also report the Chamfer distance (CD) between the point clouds of input shapes transformed by predicted and ground-truth rotations. Finally, we report the Chamfer accuracy (CA) proposed in Huang et al. [4], which measures the percentage of examples with Chamfer distance smaller than a threshold (set to 0.05).

**Results.** Quantitative results are shown in Table 3. Both Xiao et al. [44] and SoftRas obtain performance that is above chance by a large margin, while there is plenty of room for improvement. SoftRas achieves slightly better performance on all metrics using only the masks. Computational-wise, because multiple optimizations are performed during the inference, SoftRas is also slower by orders of magnitude than Xiao et al. [44].

### 4.4 Part Assembly

The task of part assembly aims to infer the assembled shapes from canonical 3D assembly parts. Specifically, given a list of parts in point clouds, the goal is to predict a 3D pose for each part such that all parts in their predicted poses constitute a reasonable final shape.

**Setup.** To construct the test dataset, we sample point clouds from assembly parts of the chair objects in IKEA-Manual the same way as in [4]. We only use the chair category because our baseline models are trained on the chair category in PartNet [9].

| | PartNet [9] (Chair) | | IKEA-Manual (Chair) | |
|---|---|---|---|---|
| | Shape CD ↓ | Part Accuracy ↑ | Shape CD ↓ | Part Accuracy ↑ |
| B-LSTM [4] | 0.0131 | 21.77 | 0.0181 | 3.48 |
| B-Global [4] | 0.0146 | 15.70 | 0.0195 | 0.87 |
| RGL-Net [20] | N/A | N/A | 0.0583 | 2.01 |
| RGL-Net [20]* | **0.0087** | **49.06** | 0.0508 | 3.99 |
| Huang et al. [4] | 0.0091 | 39.96 | **0.0151** | **6.90** |

Table 4: Quantitative results of RGL-Net [20] and Huang et al. [4] and the baseline methods in Huang et al. [4], evaluated on the chair category of PartNet and of IKEA-Manual. RGL-Net* assumes the input parts are ordered in top-bottom order. Results on PartNet are from the original papers. There is a significant performance drop from PartNet to IKEA-Manual, indicating that these models fail to generalize to novel part decomposition.

We evaluate four baselines designed for the part assembly task. The first baseline is Huang et al. [4], a graph learning method which sequentially refines predicted poses with a graph neural network backbone. The second baseline is RGL-Net [20], a graph learning method that predicts poses of parts with a recurrent framework to account for previously assembled parts. We also evaluated B-Global and B-LSTM that serve as baselines in Huang et al. [4].

We evaluate the checkpoints of these two baselines pretrained on the PartNet chair category. We report the Chamfer distance (CD) between the assembled shape and the ground truth shape. We also report Part Accuracy proposed in [4], which measures the percentage of parts that are within a fixed CD threshold (set to 0.01) from the ground truth parts.

**Results.** Quantitative results are shown in Table 4. Compared to the original results reported in both baselines, the shape CD and Part Accuracy degrade as expected. Note that for both methods, part accuracy degrades by a much larger margin compared to the shape CD. We hypothesize that this is because most parts in PartNet are simplified versions of the real-world parts and have less variety. Therefore when evaluated on our dataset, as shown in Fig. 9, the baseline models can compose parts into a reasonable overall shape of chair, but are not able to identify the correct semantic meaning of each part, resulting in poor part accuracy.

## 5 Discussion

We present IKEA-Manual, a dataset for step-by-step understanding of shape assembly from 3D models and human-designed visual manuals. The dataset contains 102 IKEA objects with realistic 3D assembly parts paired with visual assembly manuals. We also parse the plain IKEA manual images into structured formats that can be readily parsed by machines. The fine-grained 2D and 3D annotations in the dataset enable various applications such as manual generation, part segmentation, pose estimation, and 3D part assembly. One limitation of our work is that we do not model connectors between assembly parts [12, 8]. Incorporating connector modeling can enable shape assembly tasks that align better with real-world scenarios. Our dataset is biased towards the "Chair" category, which reflects the imbalanced distribution of online 3D repositories. Building models that can robustly generalize across categories is also an interesting future direction.

As our dataset can be leveraged to develop manual generation methods, related human design jobs may be fully automated or reduced. We will make it clear on the website that our dataset and tools should only be used with good intentions following appropriate licenses.

**Acknowledgements.** We thank Stephen Tian and Hong-Xing (Koven) Yu for providing detailed feedback on the paper. This work is in part supported by Autodesk, the Stanford Institute for Human-Centered Artificial Intelligence (HAI), Center for Integrated Facility Engineering (CIFE), NSF RI #2211258, NSF CCRI #2120095, ONR MURI N00014-22-1-2740, and Analog, IBM, JPMC, Salesforce, and Samsung.

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
