# OpenReview forum: "IKEA-Manual: Seeing Shape Assembly Step by Step"
_NeurIPS.cc/2022/Track/Datasets_and_Benchmarks — NeurIPS 2022 Datasets and Benchmarks _

### Official Review · Reviewer_mjCn · 2022-07-16
**Novel idea, but small dataset size**

**Rating:** 6
**Confidence:** 3

**Strengths:**

- [S1] They propose a novel task that is close to real world scenarios. In addition, their data provide both step- and part-level annotations, which allow evaluating on various tasks.
- [S2] Inspired by natural language processing, they provide appropriate evaluation protocols to evaluate with. Although the idea is novel, there are some drawbacks. I’ve written the expected weakness of the proposed metrics on [W4].
- [S3] "[C2] If symmetric components are in furnitures, multiple assembly strategies could exists. Regarding this issue, IKEA-manual provides annotations for geometrically equivalent objects." - This is a great breakthrough for the ambiguity issue on the shape assembly task.
- [S4]The attached demo video facilitated my understanding. Thanks :)

**Weaknesses:**

- [W1] The proposed dataset collection stage strongly relies on external 3D datasets. Moreover, they exclude several objects when the acquired 3D shape does not fit in manuals. Consequently, the dataset only has 102 objects and 1056 part masks, indicating that large-scale models can easily get overfitted. As visualized in Fig8, it can train a basic network like PointNet. However, larger models can be easily overfitted due to a lack of data.
- [W2] There are ambiguities of orders while assembling parts. If we used manuals as GT orders, we could ignore variations of feasible orders. If we use all possible orders as GT, we can ignore geometries. In detail, some parts cannot be assembled since the already assembled parts can interrupt the assembling step. However, this is not a major concern since most fragments have simple geometries.
- [W3] They opened possibilities for many tasks on their dataset, such as single-view 3D reconstruction and pose estimation. However, there are not sufficiently many benchmarks to verify their claims. They should provide more evaluations of baseline methods.
- [W4] The metric used for evaluating SingleStep and GeoCluster has a trivial solution(SingleStep achieves 100% precision, although its recall is low). This makes the authors to be confused since comparison can be objective.

**Additional Feedback:**

My major concern about this paper is the size of dataset.
Compared to recent 3D datasets, the size of dataset is too small so that neural networks can be easily overfitted on the proposed dataset.
In addition, depending on how users split train/val/test set, the performance can vary, involving inaccurate evaluation.
I would like to raise the score if there were more benchmarks to verify whether this dataset can be a great choice for both handcrafted and learning-based models on shape assembly tasks.

**Clarity:**

Well-written. I didn't have a hard time reading this article, although I'm not an expert in this area.

**Correctness:**

- (L101) IKEA-Manual can be used for building and evaluating models that parse assembly manuals.
    - The sentence above can be overclaimed since manually labeling building-scale data is impractical in practice due to costly manual efforts for labeling them. You should provide a somewhat automatic labeling process to ensure this potential.
- (L250) Motivated by the observation that geometrically similar parts are prone to be assembled in the same steps, we implement a clustering-based baseline GeoCluster.
    - I’m not really confident about this part. It would be better to show some examples and elaborate about this intuition.

**Documentation:**

My major concern about this paper is the size of the dataset. Compared to recent 3D datasets, the size of the dataset is too small, so neural networks can be easily overfitted on the dataset. In addition, depending on how the author split the train/val/test set, the performance can vary, indicating that evaluation of this data can be inaccurate. I want to raise the score if more benchmarks verify whether this dataset can be a great choice for both handcrafted and learning-based models on shape assembly tasks.

**Ethics:**

No ethical concerns about this data.

**Relation To Prior Work:**

They have clearly discussed about relation to prior work.

**Summary And Contributions:**

- [C1] IKEA-manual provides part-level annotations and manuals, parsed in machine-interpretable form for shape assembly. It composes of data, aligned with real-world IKEA furnitures, to allow evaluating on practical scenarios.
- [C2] If symmetric components are in furnitures, multiple assembly strategies could exist. Regarding this issue, IKEA-manual provides annotations for geometrically equivalent objects.
- [C3] They have proposed basic level of benchmarks for each task.

---

> ### Author Response · Authors · 2022-08-19
> **Response to Reviewer mjCn**
>
> Thanks for your thoughtful review and helpful suggestions!
>
> **Q1: Dataset scale.**
>
> A1: We want to clarify that the main goal of our dataset is not for large-scale training, though it is certainly an interesting future direction; instead, leveraging high-quality annotations on the alignment between real-world objects and corresponding visual manuals, IKEA-Manual serves as a test set to benchmark various tasks related to shape assembly, such as manual plan generation and part assembly. Previous datasets cannot achieve these goals because they either only contain synthetic shapes, or have no paired manuals and fine-grained annotations.
>
> Datasets of a similar scale are common in regimes where real-world data and annotations are hard to obtain. The IKEA Furniture Environment [1] has 60 objects, and the IKEA Object state dataset [2] only includes 5 objects. In other domains, the MIT intrinsics [3] and the Middlebury [4] datasets only have 16 and 33 instances, respectively, but they have greatly advanced the field of intrinsic composition and stereo vision due to the high-quality annotations. We hope that IKEA-Manual will serve a similar purpose.
>
> **Q2: Ambiguities of orders while assembling parts.**
>
> A2: We clarify that for the assembly plan generation task, we are interested in generating assembly plans that suit humans better.  Although there might be multiple feasible assembly plans for an object, humans will have preferences over one another for numerous factors like clarity and understandability [5]. Even for objects with simple geometries, humans will prefer specific assembly plans over others. For example, to assemble the chair in Fig. 7, designers choose to assemble the legs in an individual step, instead of assembling all the parts in a single step. This makes the manual plan easier to follow. Designing an automatic metric that captures the human factors is still an open problem. Therefore, we use plans from professionally designed IKEA-Manuals as a proxy for human-oriented assembly plans.
>
> **Q3: The precision metric in manual plan generation.**
>
> A3: We argue that all the metrics should be considered comprehensively for a holistic evaluation of different models. SingleStep achieves 100% precision for simple matching because it does not output any nodes apart from the root node: it assembles all the parts in one step for every IKEA object. This trivial solution will lead to a lower score for the recall metrics and all the metrics for hard matching, according to Table 2.
>
> **Q4: Goal of IKEA-Manual (L101).**
>
> A4: We clarify that our dataset is not intended for large-scale training. We have removed the word “building” in L101 for clarification.
>
> **Q5: Intuition of GeoCluster.**
>
> A5: As suggested by previous research [5], people prefer to assemble geometrically similar parts in one step, which is also empirically observed in our dataset. For example, for the chair and table objects in Figure 7, chair legs are assembled in one step, and pieces of the table top are also assembled in one step. This is not a rule that always holds, but can be used as a heuristic to build our baseline.
>
> **Q6: Split of the dataset.**
>
> A6: For the manual plan generation and part assembly tasks, all examples in IKEA-Manual are included in the test set (i.e., we are only using IKEA-Manual for evaluation purposes). For the manual segmentation task, we randomly split the data into train and test sets, and the split is included in the metadata. This ensures that future works using our dataset can have a consistent evaluation and comparison. How to split the dataset for other tasks depends on the specific formulation of tasks.

---

> > ### Author Response · Authors · 2022-08-19
> > **Response to Reviewer mjCn (Cont')**
> >
> > **Q7: More benchmarks and baselines.**
> >
> > A7: To further demonstrate the broad applicability of our dataset, we have included another benchmarking task, part-conditioned pose estimation built on IKEA-Manual. Given a 3D part and a manual image of its 2D projection, the model needs to predict the part's rotation. We selected three baselines for this task: 1. a model that randomly predicts rotations regardless of the input; 2. a learning-based model proposed in Xiao et al. [6]; 3. an analysis-by-synthesis algorithm using SoftRas [6]. The results are
> >
> >
> > |                | Rotation Error ↓ | Rotation Accuracy ↑ | Chamfer Distance ↓ | Chamfer Accuracy ↑ | Inference Time (s) ↓ |
> > |:--------------:|:----------------:|:-------------------:|:------------------:|:------------------:|:--------------------:|
> > |     Random     |      103.49      |         3.05        |       0.1344       |        0.28        |          N/A         |
> > |   Xiao et al. [6] |       82.79      |        20.56        |       0.0984       |        49.53       |        <10e-5        |
> > | SoftRas [7] |       64.81      |        24.30        |       0.0976       |        51.4        |          243         |
> >
> > Both Xiao et al. [6] and SoftRas obtain performance that beats the random baseline by a large margin, but there is still plenty of room for improvement. SoftRas achieves slightly better performance on all metrics, but is also significantly slower due to the optimization process during inference. We have included more details in Section 4.3.
> >
> > We also included two extra baselines, B-Global and B-LSTM [8], for the part assembly task. The results are
> >
> > |              |      PartNet     |               |    IKEA-Manual   |               |
> > |:------------:|:----------------:|:-------------:|:----------------:|:-------------:|
> > |              | Chamfer Distance | Part Accuracy | Chamfer Distance | Part Accuracy |
> > |   B-Global   |      0.0146      |     15.70     |      0.0195      |      0.87     |
> > |    B-LSTM    |      0.0131      |     21.77     |      0.0181      |      3.48     |
> > |    RGL-Net   |      0.0087      |     49.06     |      0.0583      |      2.01    |
> > | PartAssembly |      0.0091      |     39.96     |      0.0151      |      6.90     |
> >
> > We can see that none of the baselines perform well for this task, suggesting that our IKEA-Manual dataset remains a challenge for existing AI algorithms.
> >
> > Again, thank you for your comments. We hope that our responses address your concerns. Feel free to let us know if you have any additional questions.
> >
> >
> > [1] Youngwoon Lee, Edward S. Hu, and Joseph J. Lim. IKEA furniture assembly environment for long-horizon complex manipulation tasks. In ICRA 2021.
> >
> > [2] Yongzhi Su, Mingxin Liu, Jason Rambach, Antonia Pehrson, Anton Berg, and Didier Stricker. "IKEA Object State Dataset: A 6DoF object pose estimation dataset and benchmark for multi-state assembly objects. In arXiv 2021.
> >
> > [3] Roger Grosse, Micah K. Johnson, Edward H. Adelson, and William T. Freeman, Ground truth dataset and baseline evaluations for intrinsic image algorithms. In ICCV 2009.
> >
> > [4] Scharstein, Daniel, Heiko Hirschmüller, York Kitajima, Greg Krathwohl, Nera Nešić, Xi Wang, and Porter Westling. "High-resolution stereo datasets with subpixel-accurate ground truth." In GCPR 2014.
> >
> > [5] Maneesh Agrawala, Doantam Phan, Julie Heiser, John Haymaker, Jeff Klingner, Pat Hanrahan, and Barbara Tversky. Designing effective step-by-step assembly instructions. In Transactions on Graphics 2003.
> >
> > [6] Xiao, Yang, Xuchong Qiu, Pierre-Alain Langlois, Mathieu Aubry, Renaud Marlet, and France Champs-sur-Marne. Pose from Shape: Deep Pose Estimation for Arbitrary 3D Objects. In BMVC 2019.
> >
> > [7] Shichen Liu, Tianye Li, Weikai Chen, and Hao Li. Soft rasterizer: A differentiable renderer for image-based 3d reasoning. In ICCV 2019.
> >
> > [8] Guanqi Zhan, Qingnan Fan, Kaichun Mo, Lin Shao, Baoquan Chen, Leonidas J. Guibas, and Hao Dong. Generative 3d part assembly via dynamic graph learning. In NeurIPS 2020.

---

> > > ### Author Response · Authors · 2022-08-26
> > > **Looking Forward to Your Feedback**
> > >
> > > Dear Reviewer,
> > >
> > > Thank you again for the constructive reviews, which have helped us improve the quality and clarity of our paper. In our revision, we have included a new pose estimation task with three baselines. We have also added two baselines for the part assembly task. The manuscript is updated to clarify the motivation of our evaluation for manual plan generation.  We hope our response and new results can address your concerns. As we approach the end of the discussion period, please don’t hesitate to let us know if you have any additional questions or comments!
> > >
> > > Thanks for your time,
> > >
> > > Authors

---

> > > > ### Comment · Reviewer_mjCn · 2022-08-29
> > > > **I agree most of them but still not convinced about the order ambiguity.**
> > > >
> > > > Thanks for your replies.
> > > >
> > > > Most replies are convincing but still the order ambiguity is not clear for me and this seems quite challenging to handle with the proposed idea. In my opinion, although human-centric preference could exist, a different order should not be handled as a wrong answer. Since many manuals are drawn from various designers, this ambiguity should be solved with different metrics such as graph similarity metrics.
> > > >
> > > > But I'm sure that this could be future work and the reasons why I first put my score to WR are addressed. Thanks. I've raised my score to WA.

---

> > > > > ### Author Response · Authors · 2022-08-29
> > > > > **Thank You**
> > > > >
> > > > > Dear Reviewer mjCn,
> > > > >
> > > > > We agree that developing metrics that can better measure human-centric preference is a challenging but important future direction. We will include more discussions about this point in the paper.
> > > > >
> > > > > We would like to thank you again for your constructive review. We are happy to see that our response and revision have addressed your concerns. We sincerely appreciate your suggestions.
> > > > >
> > > > > Thanks for your time,
> > > > >
> > > > > Authors

---

### Official Review · Reviewer_ZVVh · 2022-07-19
**IKEA-Manual: Seeing Shape Assembly Step by Step**

**Rating:** 7
**Confidence:** 3
**Clarity:** The paper is well written and its rea…

**Strengths:**

- The authors performed a thorough comparison  of the proposed dataset with respect to existing ones on different axes.
- The authors formalized and described problems that could be easily addressed using the propsed dataset.
- Out of the different annotations performed, the 2D-3D alignment between 3D parts and manuals images is certainly the most important one as it alllows to create the bridge between 3D shapes and 2D manuals. Only Pix3D dataset was offering this type of annotations previously.


**Weaknesses:**

One limitation of the dataset resides in the fact that  connectors like screws and hinges, which are used to fasten the object parts in the assembly manuals, are ignored similar to existing datasets. This prevents the dataset to be used for real-world scenarios. That being said, the authors are well aware of this limitation.

**Additional Feedback:**

very well written paper

**Correctness:**

The authors discussed the steps involved in the creation of the dataset from  the data collection process to the annotation process. On top of that, experiment protocols are clearly discussed for the 3 introduced tasks that could be addressed using the proposed datasets.

**Documentation:**

The paper provides  sufficient detail on data collection and organization. The authors are planning to release the dataset on the Zenodo platform.
One thing the decumentation could have benefit from is an example of script showing how to load (i.e., play with) the different files present in the dataset for a given object.

**Ethics:**

No ethical concerns. The provided annotations of visual manuals are released under CC BY-NC-SA 4.0 while the Copyright of the original visual manuals and 3D models, which are part of the dataset, are owned by their creators respectively.

**Relation To Prior Work:**

As mentioned earlier, a thorough comparison is performed with respect to existing datasets on multiple axes

**Summary And Contributions:**

In response to a lack of realistic 3D assembly objects that have paired manuals as well as the
difficulty of extracting structured information from purely image-based manuals, the authors proposed a dataset consisting of 102 IKEA objects paired with assembly manuals. They collected 3D objects of IKEA furniture from existing datasets and online repositories, paired with corresponding manuals from IKEA’s official website. For each object, they decomposed its 3D model into assembly parts that match the assembly manual and annotate their connection relationships. They annotated assembly manuals by first recording which parts are connected in each step, and then manually segmenting out the projection of parts on the manual images. Finally, they annotated keypoints on the 3D assembly parts and their 2D projections on the manual image, and used them to compute 3D poses of parts and also to build 2D-3D correspondences.

They also conducted initial experiments to illustrate the usefulness of the proposed dataset.

---

> ### Author Response · Authors · 2022-08-19
> **Response to Reviewer ZVVh**
>
> Thanks for your thoughtful review and helpful suggestions!
>
> As discussed in Section 5, we agree that modeling connectors like screws and hinges is important for building models that better align with real-world scenarios. Connectors are challenging to model and typically absent in the 3D shapes from online repositories, including many of the 3D models in IKEA-Manual. Therefore, in this paper, we focused on connections between geometric parts, which is also a common assumption in previous shape assembly works [1,2]. We believe that modeling connectors in shape assembly tasks is an exciting future direction. We hope that our responses address your concerns. Feel free to let us know if you have any additional questions.
>
> [1] Youngwoon Lee, Edward S. Hu, and Joseph J. Lim. IKEA furniture assembly environment for long-horizon complex manipulation tasks. In ICRA 2021.
>
> [2] Yichen Li, Kaichun Mo, Lin Shao, Minhyuk Sung, and Leonidas Guibas. Learning 3d part assembly from a single image. In ECCV 2020.

---

> > ### Author Response · Authors · 2022-08-26
> > **Looking Forward to Your Feedback**
> >
> > Dear Reviewer,
> >
> > Thank you again for the constructive reviews, which have helped us improve the quality and clarity of our paper. We hope our response can address your concerns. As we approach the end of the discussion period, please don’t hesitate to let us know if you have any additional questions or comments!
> >
> > Thanks for your time,
> >
> > Authors

---

### Official Review · Reviewer_gidi · 2022-07-27

**Rating:** 5
**Confidence:** 5
**Correctness:** The dataset construction is technical…
**Clarity:** Yes.

**Strengths:**

1. This dataset is very interesting for researchers who are studying shape assembly and/or planning.

2. The motivation of the dataset is very clear.

3. The paper reads well.

**Weaknesses:**

1. Table 1 should include AutoMate [18].

2. The number of existing methods for benchmarking is 2, which is relatively small. I think it is important to include all baseline methods considered in Huang et al.'s paper [4] for performance benchmarking.

3. The number of objects is only 102. Given this small number, I am guessing the performance reported in Table 3 for IKEA-Manual being this bad compared to those for PartNet might be due in part to the number of data used to train the models. I think it is important to scale the size of the IKEA-Manual dataset up to the size comparable to PartNet and then benchmark performance on this larger dataset.

4. This dataset is based on the availability of 3D models from IKEA as pointed out by the authors. If IKEA no longer authorizes those 3D models (say IKEA changes the license) or even removes them, then no one will have access to this dataset. Given this, I think the authors should come up with a detailed plan on how to deal with such potential issues. It is very important that this dataset is always available to every researcher who is interested in this topic.

**Additional Feedback:**

Please see comments in the weaknesses section above.

**Documentation:**

The dataset is accessible. The documentation is clear.

**Relation To Prior Work:**

Related work is adequate.

**Summary And Contributions:**

This paper proposed a dataset called IKEA-Manual. This is a dataset for the shape assembly task that includes 3D models of furniture and human-designed visual manuals. This dataset contains 102 IKEA objects. The authors benchmarked two existing shape assembly methods, RGL-Net and Huang et al. on the proposed dataset.

---

> ### Author Response · Authors · 2022-08-19
> **Author Response to Reviewer gidi**
>
> Thanks for your thoughtful review and helpful suggestions!
>
> **Q1: AutoMate.**
>
> A1: Thanks for the suggestion. We have included AutoMate in Table 1.
>
> **Q2: More baseline methods.**
>
> A2: We included two extra baselines, B-Global and B-LSTM [1] for the part assembly task. We did not include B-complement because the training data are not released in the official repo. The results are
>
> |              |      PartNet     |               |    IKEA-Manual   |               |
> |:------------:|:----------------:|:-------------:|:----------------:|:-------------:|
> |              | Chamfer Distance | Part Accuracy | Chamfer Distance | Part Accuracy |
> |   B-Global   |      0.0146      |     15.70     |      0.0195      |      0.87     |
> |    B-LSTM    |      0.0131      |     21.77     |      0.0181      |      3.48     |
> |    RGL-Net   |      0.0087      |     49.06     |      0.0583      |      2.01     |
> | PartAssembly |      0.0091      |     39.96     |      0.0151      |      6.90     |
>
> We can see that none of the baselines perform well for this task, suggesting that our IKEA-Manual dataset remains a challenge for existing AI algorithms.
>
> To further demonstrate the broad applicability of our dataset, we have also included another benchmarking task, part-conditioned pose estimation built on IKEA-Manual. Given a 3D part and a manual image of its 2D projection, the model needs to predict the part's rotation. We selected three baselines for this task: 1. a model that randomly predicts rotations regardless of the input; 2. a learning-based model proposed in Xiao et al. [2]; 3. an analysis-by-synthesis algorithm using SoftRas [3]. The results are
>
>
> |                | Rotation Error ↓ | Rotation Accuracy ↑ | Chamfer Distance ↓ | Chamfer Accuracy ↑ | Inference Time (s) ↓ |
> |:--------------:|:----------------:|:-------------------:|:------------------:|:------------------:|:--------------------:|
> |     Random     |      103.49      |         3.05        |       0.1344       |        0.28        |          N/A         |
> |   Xiao et al. [2] |       82.79      |        20.56        |       0.0984       |        49.53       |        <10e-5        |
> | SoftRas [3] |       64.81      |        24.30        |       0.0976       |        51.4        |          243         |
>
> Both Xiao et al. [2] and SoftRas obtain performance that beats the random baseline by a large margin, but there is still plenty of room for improvement. SoftRas achieves slightly better performance on all metrics, but is also significantly slower due to the optimization process during inference. We have included more details in Section 4.3.
>
>
>
> **Q3: Result of Part Assembly in Table 3.**
>
> A2:  We clarify that for the experiment of the part assembly task, we use IKEA-Manual to evaluate the models pretrained on PartNet. We did not use IKEA-Manual to train the models. In this experiment, we show that models trained on synthetic shapes with only semantic decomposition from PartNet cannot generalize to predict the assembly of shapes in our dataset, which contain 3D parts that are aligned with real-world settings.
>
> **Q4: Scaling to the size of PartNet.**
>
> A1: We want to clarify that different from PartNet, the main goal of our dataset is not for large-scale training, though it is certainly an interesting future direction; instead, leveraging high-quality annotations on the alignment between real-world objects and corresponding visual manuals, IKEA-Manual serves as a test set to benchmark various tasks related to shape assembly, such as manual plan generation and part segmentation. PartNet cannot achieve these goals because shapes in PartNet do not have paired manuals and fine-grained annotations related to manual-guided shape assembly.
>
> Datasets of a similar scale are common in regimes where real-world data and annotations are hard to obtain. The IKEA Furniture Environment [4] has 60 objects, and the IKEA Object state dataset [5] only includes 5 objects. In other domains, the MIT intrinsics [6] and the Middlebury [7] datasets only have 16 and 33 instances, respectively, but they have greatly advanced the field of intrinsic composition and stereo vision due to the high-quality annotations. We hope that IKEA-Manual will serve a similar purpose.
>
> **Q5: Availability of the 3D models.**
>
> A5: We clarify that only 3 out of 102 objects in IKEA-Manual are from the IKEA Object State dataset. They are available under CC BY-NC-SA 4.0. All other models are not from IKEA, but from existing public datasets and online repositories (except for one object, for which we explicitly obtain approval from the creator). This follows the practice of widely used datasets, including ShapeNet and PartNet. We will regularly maintain the availability of these models in our dataset as ShapeNet does.
>
> Again, thank you for your comments. We hope that our responses address your concerns. Feel free to let us know if you have any additional questions.

---

> > ### Author Response · Authors · 2022-08-19
> > **Author Response to Reviewer gidi (Cont')**
> >
> > [1] Guanqi Zhan, Qingnan Fan, Kaichun Mo, Lin Shao, Baoquan Chen, Leonidas J. Guibas, and Hao Dong. Generative 3d part assembly via dynamic graph learning. In NeurIPS 2020.
> >
> > [2] Xiao, Yang, Xuchong Qiu, Pierre-Alain Langlois, Mathieu Aubry, Renaud Marlet, and France Champs-sur-Marne. Pose from Shape: Deep Pose Estimation for Arbitrary 3D Objects. In BMVC 2019.
> >
> > [3] Shichen Liu, Tianye Li, Weikai Chen, and Hao Li. Soft rasterizer: A differentiable renderer for image-based 3d reasoning. In ICCV 2019.
> >
> > [4] Youngwoon Lee, Edward S. Hu, and Joseph J. Lim. IKEA furniture assembly environment for long-horizon complex manipulation tasks. In ICRA 2021.
> >
> > [5] Yongzhi Su, Mingxin Liu, Jason Rambach, Antonia Pehrson, Anton Berg, and Didier Stricker. "IKEA Object State Dataset: A 6DoF object pose estimation dataset and benchmark for multi-state assembly objects. In arXiv 2021.
> >
> > [6] Roger Grosse, Micah K. Johnson, Edward H. Adelson, and William T. Freeman, Ground truth dataset and baseline evaluations for intrinsic image algorithms. In ICCV 2009.
> >
> > [7] Scharstein, Daniel, Heiko Hirschmüller, York Kitajima, Greg Krathwohl, Nera Nešić, Xi Wang, and Porter Westling. "High-resolution stereo datasets with subpixel-accurate ground truth." In GCPR 2014.

---

> > ### Comment · Reviewer_gidi · 2022-08-25
> > **Further clarifications**
> >
> > Thank you for addressing the comments. RGL-Net performs the best on PartNet, but the worst on IKEA-Manual. Can the authors explain why this is happening? I think more analysis on why these methods don't work well should be provided. This will help or even motivate people to develop novel methods for this dataset.

---

> > > ### Author Response · Authors · 2022-08-27
> > > **Follow Up**
> > >
> > > **Q1: Gap between results on PartNet and IKEA-Manual for RGL-Net.**
> > >
> > > A1: Thank you for the suggestion! We think the primary reason why RGL-Net performs significantly better on PartNet than IKEA-Manual is that RGL-Net relies on a "privileged" assumption of input data on PartNet. More specifically, RGL-Net assumes that all input parts are sorted by their height (from top to bottom) as in the **groundtruth** final shape. That is, for example, for chairs, the original paper assumes that in the input sequence, the chair back comes first, then the seat and legs. Such ordering assumption provides additional information such as relative heights of parts and semantic groupings (e.g., chair legs are ordered next to each other in the input sequence). This is an **unrealistic** assumption because it implicitly assumes access to the final shape. Therefore, in IKEA-Manual, we do not provide such an order in inputs. We will clarify this in the paper.
> > >
> > > To illustrate the significance of this assumption on model performance, we add the following experiment. In IKEA-Manual, we adopt the same top-down input order by sorting parts by the height of their center in the final shape (computed as the center of the 3D bounding box). The results are
> > >
> > > |   Dataset   | Input Order | Shape CD ↓ | Part Accuracy ↑ |
> > > |:-----------:|:-----------:|:----------:|:---------------:|
> > > |   PartNet   |   Top-Down  |   0.0087   |      49.06      |
> > > | IKEA-Manual |   Top-Down  |   0.0508   |       3.99      |
> > > | IKEA-Manual |    Random   |   0.0583   |       2.01      |
> > >
> > > Compared to the random order, this model performs better, beating the Bi-Global and Bi-LSTM model baselines on the part accuracy metric. But it is still worse than PartAssembly (Huang et al). A possible reason is that the order information for IKEA-Manual is not as informative as that in PartNet, since the parts usually do not correspond to semantic classes.
> > >
> > > **Q2: Poor performance of baseline models.**
> > >
> > > We think the baseline models do not work well on IKEA-Manual mainly because there is a gap between the distribution of the parts in PartNet and that of IKEA-Manual. In this figure ([link](https://download.cs.stanford.edu/viscam/ikea_manual/parts_visualization.jpg)), we illustrate this gap by visualizing 50 parts randomly sampled from the chair category of PartNet and IKEA-Manual datasets respectively. From the figure, we can see the parts in PartNet are typically simple geometric primitives such as bars and planes, while parts in IKEA-Manual have more complex geometric structures that align with real-world assembly scenarios. We believe developing methods that are more robust to this type of geometric domain shift is critical for tackling the part assembly task on IKEA-Manual. We will include the figure and discussion in our paper.
> > >
> > > Thanks for your feedback. We sincerely hope our reply has better addressed your concerns. Feel free to let us know if you have any additional questions!

---

### Official Review · Reviewer_Z7NU · 2022-07-28

**Rating:** 6
**Confidence:** 3
**Correctness:** Yes.
**Clarity:** Yes.

**Strengths:**

The most significant strength of the paper is all objects are real objects (IKEA furnitures) instead of synthetic ones. Most works in this area focus on the assembly of synthetic objects since it's hard to collect the data of real object parts. Any efforts on real data of part assembly is an important contribution to our community.

**Weaknesses:**

My primary concern is the scale of the IKEA-Manual dataset.

- The dataset only has 102 objects. Considering that the scale is relatively small, it may not suitable for large-scale training.
- It might be challenging to scale to more objects: (1) the number of available IKEA manuals on the IKEA website limits the number of objects, which does not allow it to scale; (2) the manual annotation is also very labor-intensive. It seems challenging to scale to more objects. How long does it need to annotate a single object?
- I also have ethics concerns. See ethics.

**Additional Feedback:**

Do all objects also come with RGB image besides depth? Pix3D object should have RGB but the paper does not seem to mention RGB anywhere.

**Documentation:**

Yes.

**Ethics:**

I'm worried about whether this is allowed given that all annotations are based on IKEA's official manual, whose copyright definitely belongs to IKEA.

**Relation To Prior Work:**

Yes.

**Summary And Contributions:**

The paper proposes IKEA-Manual dataset which consists of 102 IKEA objects paired with parts and assembly manuals. The original IKEA manual is collected on the IKEA website, while objects models are available in various datasets such as IKEA and Pix3D. They are then annotated to build the shape assembly procedure. The dataset has a broad application including assembly plan generation, part segmentation and 3D part assembly.

---

> ### Author Response · Authors · 2022-08-19
> **Author Response to Reviewer Z7NU**
>
> Thanks for your thoughtful review and helpful suggestions!
>
> **Q1: Dataset scale.**
>
> A1: We want to clarify that the main goal of our dataset is not for large-scale training, though it is certainly an interesting future direction; instead, leveraging high-quality annotations on the alignment between real-world objects and corresponding visual manuals, IKEA-Manual serves as a test set to benchmark various tasks related to shape assembly, such as manual plan generation and part assembly. Previous datasets cannot achieve these goals because they either only contain synthetic shapes, or have no paired manuals and fine-grained annotations.
>
> Datasets of a similar scale are common in regimes where real-world data and annotations are hard to obtain. The IKEA Furniture Environment [1] has 60 objects, and the IKEA Object state dataset [2] only includes 5 objects. In other domains, the MIT intrinsics [3] and the Middlebury [4] datasets only have 16 and 33 instances, respectively, but they have greatly advanced the field of intrinsic composition and stereo vision due to the high-quality annotations. We hope that IKEA-Manual will serve a similar purpose.
>
> **Q2: The annotation time for a single object.**
>
> A2: The annotation time for a single object is dependent on the number of parts and complexity of the manual for each object. This ranges from 2 hours to 5 hours.
>
> **Q3: Are annotations allowed?.**
>
> A3: Yes, the annotations are allowed and there is no ethical concern. In fact, we have been communicating closely with the IKEA Digital Lab about the development of the dataset. The IKEA Digital Lab, which introduced the Object State Dataset, is aware of and supports our efforts. There is also the possibility of a more extensive collaboration in the longer-term future, if our dataset is widely adopted by the community.
>
> **Q4: RGB image of the IKEA objects.**
>
> A4: We clarify that there are no RGB or depth images in our dataset. The main focus of our dataset is on tasks related to manual-guided shape assembly. Our dataset contains visual manual images of line drawing styles, as well as their segmentation and correspondence with the CAD models, as illustrated in Figure 1. Using drawing instead of RGB or depth images is a common design choice for real-world manuals, such as IKEA and LEGO.
>
> Again, thank you for your comments. We hope that our responses address your concerns. Feel free to let us know if you have any additional questions.
>
> [1] Youngwoon Lee, Edward S. Hu, and Joseph J. Lim. IKEA furniture assembly environment for long-horizon complex manipulation tasks. In ICRA 2021.
>
> [2] Yongzhi Su, Mingxin Liu, Jason Rambach, Antonia Pehrson, Anton Berg, and Didier Stricker. "IKEA Object State Dataset: A 6DoF object pose estimation dataset and benchmark for multi-state assembly objects. In arXiv 2021.
>
> [3] Roger Grosse, Micah K. Johnson, Edward H. Adelson, and William T. Freeman, Ground truth dataset and baseline evaluations for intrinsic image algorithms. In ICCV 2009.
>
> [4] Scharstein, Daniel, Heiko Hirschmüller, York Kitajima, Greg Krathwohl, Nera Nešić, Xi Wang, and Porter Westling. "High-resolution stereo datasets with subpixel-accurate ground truth." In GCPR 2014.

---

> > ### Author Response · Authors · 2022-08-26
> > **Looking Forward to Your Feedback**
> >
> > Dear Reviewer,
> >
> > Thank you again for the constructive reviews, which have helped us improve the quality and clarity of our paper. We hope our response can address your concerns. As we approach the end of the discussion period, please don’t hesitate to let us know if you have any additional questions or comments!
> >
> > Thanks for your time,
> >
> > Authors

---

> > ### Comment · Reviewer_Z7NU · 2022-08-27
> > **Re: Author Response to Reviewer Z7NU**
> >
> > Thanks for your detailed response. I'm convinced for the dataset scale issue, and I'm glad to know there is no ethical concern. I'm happy to change my rating from 5 to 6.

---

> > > ### Author Response · Authors · 2022-08-29
> > > **Thank You**
> > >
> > > Dear Reviewer Z7NU,
> > >
> > > We would like to thank you again for your constructive review. We are happy to see that our response has addressed your concerns. We sincerely appreciate your suggestions.
> > >
> > > Thanks for your time,
> > >
> > > Authors

---

### Official Review · Reviewer_9FiH · 2022-07-28

**Rating:** 7
**Confidence:** 4

**Strengths:**

- The authors present 102 objects representing diversity, and the usefulness of the dataset relies on its direct connection with the manuals.
- Their 3D-2D connection.
- The dataset can be applied to three main tasks.
- Section 4 presents the application of the dataset formulating the problem. Section 4 also discusses criteria and metrics like simple matching and hard matching when using precision and recall for its evaluation.


**Weaknesses:**

- Because of previous IKEA work, like IKEA Object State Dataset, it could be misunderstood as an extension of another dataset.
- More extensive ablation studies could benefit the paper by showing more diverse results when using this dataset.

**Additional Feedback:**

Will the authors provide the annotation tools? The supplemental material has videos showing connection segmentation annotation and keypoint annotation, and these tools can help augment this dataset in future research. Besides the three main tasks proposed in the paper, do the authors have other tasks that can use this dataset?

**Clarity:**

- It's easy to read and follow.
- Line 154 'for' wrote twice.

**Correctness:**

It's understandable and consistent with previous work what makes the collection of the dataset useful.

**Documentation:**

supplementary material, python code website to download the dataset

**Ethics:**

No humans in the dataset. IKEA manuals are of public access.

**Relation To Prior Work:**

- More directly related to IKEA-OSD.

**Summary And Contributions:**

The paper presents IKEA-Manual, a dataset of 102 IKEA objects and their assembly manuals, mainly focused on three tasks that provide benefits directly related to their manuals.
The authors' contributions are:
- Compared to the five objects presented by IKEA OSD, IKEA-Manual presents 102 objects.
- Three tasks: assembly plan generation task, part segmentation, and 3D part assembly.
- Tree-structured annotations for the assembly process.

---

> ### Author Response · Authors · 2022-08-19
> **Author Response to Reviewer 9FiH**
>
> Thanks for your thoughtful review and helpful suggestions!
>
> **Q1: Difference with the IKEA Object State Dataset.**
>
> A1:  In Table 1, we have compared IKEA-Manual with other IKEA-related datasets, which demonstrates our dataset is significantly different from previous works. Specifically, the IKEA Object State dataset does not have annotated plans, manual segmentations, and 2D-3D correspondence between 3D parts and manuals. Moreover, they only have 5 objects in the dataset, which limits the usefulness of the dataset.
>
> **Q2: More baselines and tasks.**
>
> A2: We included two extra baselines, B-Global and B-LSTM [1] for the part assembly task. The results are
>
> |              |      PartNet     |               |    IKEA-Manual   |               |
> |:------------:|:----------------:|:-------------:|:----------------:|:-------------:|
> |              | Chamfer Distance | Part Accuracy | Chamfer Distance | Part Accuracy |
> |   B-Global   |      0.0146      |     15.70     |      0.0195      |      0.87     |
> |    B-LSTM    |      0.0131      |     21.77     |      0.0181      |      3.48     |
> |    RGL-Net   |      0.0087      |     49.06     |      0.0583      |      2.01     |
> | PartAssembly |      0.0091      |     39.96     |      0.0151     |      6.90     |
>
> We can see that none of the baselines perform well for this task, suggesting that our IKEA-Manual dataset remains a challenge for existing AI algorithms.
>
> To further demonstrate the broad applicability of our dataset, we have also included another benchmarking task, part-conditioned pose estimation built on IKEA-Manual. Given a 3D part and a manual image of its 2D projection, the model needs to predict the part's rotation. We selected three baselines for this task: 1. a model that randomly predicts rotations regardless of the input; 2. a learning-based model proposed in Xiao et al. [2]; 3. an analysis-by-synthesis algorithm using SoftRas [3]. The results are
>
>
> |                | Rotation Error ↓ | Rotation Accuracy ↑ | Chamfer Distance ↓ | Chamfer Accuracy ↑ | Inference Time (s) ↓ |
> |:--------------:|:----------------:|:-------------------:|:------------------:|:------------------:|:--------------------:|
> |     Random     |      103.49      |         3.05        |       0.1344       |        0.28        |          N/A         |
> |   Xiao et al. [2] |       82.79      |        20.56        |       0.0984       |        49.53       |        <10e-5        |
> | SoftRas [3] |       64.81      |        24.30        |       0.0976       |        51.4        |          243         |
>
> Both Xiao et al. [2] and SoftRas obtain performance that beats the random baseline by a large margin, but there is still plenty of room for improvement. SoftRas achieves slightly better performance on all metrics, but is also significantly slower due to the optimization process during inference. We have included more details in Section 4.3.
>
> **Q3: Annotation tools.**
>
> A3: We will release the annotation tools together with the dataset to benefit future research.
>
> Again, thank you for your comments. We hope that our responses address your concerns. Feel free to let us know if you have any additional questions.
>
>
> [1] Guanqi Zhan, Qingnan Fan, Kaichun Mo, Lin Shao, Baoquan Chen, Leonidas J. Guibas, and Hao Dong. Generative 3d part assembly via dynamic graph learning. In NeurIPS 2020.
>
> [2] Xiao, Yang, Xuchong Qiu, Pierre-Alain Langlois, Mathieu Aubry, Renaud Marlet, and France Champs-sur-Marne. Pose from Shape: Deep Pose Estimation for Arbitrary 3D Objects. In BMVC 2019.
>
> [3] Shichen Liu, Tianye Li, Weikai Chen, and Hao Li. Soft rasterizer: A differentiable renderer for image-based 3d reasoning. In ICCV 2019.

---

> > ### Author Response · Authors · 2022-08-26
> > **Looking Forward to Your Feedback**
> >
> > Dear Reviewer,
> >
> > Thank you again for the constructive reviews, which have helped us improve the quality and clarity of our paper. In our revision, we have added two baselines for the part assembly task. We have also included a new pose estimation task with three baselines. We hope our response and new results can address your concerns. As we approach the end of the discussion period, please don’t hesitate to let us know if you have any additional questions or comments!
> >
> > Thanks for your time,
> >
> > Authors

---

### Official Review · Reviewer_b5DC · 2022-07-28
**Interesting dataset with multiple application domains**

**Rating:** 7
**Confidence:** 3
**Correctness:** Yes, no comments here.
**Clarity:** Yes.

**Strengths:**

* The paper is well-written
* Identifies a gap in existing datasets and tackles it
* Several annotations to support multiple tasks
* Interesting and insightful choice of baselines for the addressed tasks


**Weaknesses:**

* Clearly mention in the discussion section the limitation of the datasets in terms of distribution of objects (dominantly chairs)
* Add a phrase about EPnP or expand it to Efficient Perspective-n-Point Camera Pose Estimation
* In line 269, I understood that the images are coming from the different manuals of the 102 objects. I suggest elaborating more on the split (whether images of the same object were all in one split or all images were shuffled and 40 were selected for testing)
* if possible, create a Git repository for the dataset (include some scripts to read the data, visualize examples, perform a split, etc...)



**Additional Feedback:**

The paper is well-written, and I noticed these possible minor corrections:

* Line 103: I think it is “followed by”
* Figure 3, either expanding kps to key-points or mentioning this in the caption would be helpful to readers
* Line 154: there is an extra “for”
* Line 195: there is probably an extra dot after the word “repositories”
* Line 209: the sentence starting with “Finally” is a bit confusing and can be rephrased for clarity.


**Documentation:**

Yes.

**Ethics:**

Not to my knowledge and understanding.

**Relation To Prior Work:**

Yes, clearly explained.

**Summary And Contributions:**

The paper identifies a gap in datasets with 3d assembly objects with paired manuals and introduces a dataset with 102 IKEA objects paired with assembly manuals. Additionally, they include several annotations. The paper highlights the broad applications of the dataset and illustrates tasks and respective baselines.

---

> ### Author Response · Authors · 2022-08-19
> **Author Response to Reviewer b5DC**
>
> Thanks for your thoughtful review and helpful suggestions!
>
> **Q1: Split of the part-conditioned manual segmentation task.**
>
> A1: We shuffled all the images altogether and selected 353 examples for training and 40 for testing.
>
> **Q2: Git repository.**
>
> A2: Yes, we will create a Git repository for the dataset and our annotation tool upon publication.
>
> We also corrected the typos and added a limitation discussion about the imbalanced distribution of our dataset in Section 6.
>
> Again, thank you for your comments. We hope that our responses address your concerns. Feel free to let us know if you have any additional questions.

---

> > ### Author Response · Authors · 2022-08-26
> > **Looking Forward to Your Feedback**
> >
> > Dear Reviewer,
> >
> > Thank you again for the constructive reviews, which have helped us improve the quality and clarity of our paper. We hope our response can address your concerns. In our revision, we have included a discussion of category imbalance in Section 5 and fixed several typos. As we approach the end of the discussion period, please don’t hesitate to let us know if you have any additional questions or comments!
> >
> > Thanks for your time,
> >
> > Authors

---

### Author Response · Authors · 2022-08-19
**Summary of Revisions**

We thank all reviewers for their thoughtful comments and helpful suggestions. We have made revisions to our paper accordingly, which are highlighted in blue in the updated submission. The main changes include

### Experiments:
- In Section 4.3, we have introduced a new benchmarking task called part-conditioned pose estimation built on IKEA-Manual. We have evaluated three baselines for this task. The results suggest that current models still have a large room for improvement for this task.
- In Section 4.4, we have included two more baselines for the part assembly task. None of the baselines perform well for this task, suggesting that our IKEA-Manual dataset remains a challenge for existing AI algorithms.
### Writing:
- We have clarified the motivation of our evaluation for manual plan generation in Section 4.1.
- We have included a discussion of category imbalance in Section 5.


We further address questions from each reviewer in the individual responses. Feel free to let us know if you have any additional questions!

---

### Meta-Review · Area_Chair_VcDy · 2022-09-10

**Recommendation:** Accept
**Confidence:** 5

**Metareview:**

This submission proposes an interesting benchmark dataset with fine-grained annotations for 3D IKEA object assembly. Many of the concerns the reviewers raised are well addressed by the authors in discussion, and it obtains fairly positive concerns at the end. AC finds that the small scale of the dataset and the order ambiguity of annotations are weak points indeed but this dataset addresses several limitations of previous related datasets, facilitating several potential tasks for future research in the community. Therefore, AC recommends accepting this work.

---

### Decision · Program_Chairs · 2022-09-16

Accept